# Gestational Diabetes: Overview with Emphasis on Medical Management

**DOI:** 10.3390/ijerph17249573

**Published:** 2020-12-21

**Authors:** Michelle Lende, Asha Rijhsinghani

**Affiliations:** Division of Maternal and Fetal Medicine, Department of Obstetrics and Gynecology, Albany Medical Center, Albany Medical College, Albany, NY 12208, USA; lendem@amc.edu

**Keywords:** gestational diabetes

## Abstract

With the rising trend in obesity, the incidence of gestational diabetes mellitus (GDM) and perinatal complications associated with the condition are also on the rise. Since the early 1900s, much knowledge has been gained about the diagnosis, implications, and management of gestational diabetes with improved outcomes for the mother and fetus. Worldwide, there is variation in the definition of GDM, methods to screen for the condition, and management options. The International Association of Diabetes in Pregnancy Study Groups has published recommendations for a one-step approach to screen pregnant women for GDM, in order to develop outcome-based criteria that can be used internationally. However, management of GDM continues to be varied, and currently several options are available for treatment of hyperglycemia during pregnancy. A review of various aspects of GDM is discussed with a focus on the medical management during pregnancy, as practiced in the United States.

## 1. Introduction

A worldwide rising trend in obesity has been reported from 1975 to 2016, affecting females and males alike [1]. In women, the rising obesity has led to an increase in the incidence of gestational diabetes mellitus (GDM) as well as associated pregnancy and perinatal complications. Known non-modifiable risk factors for predisposition to GDM include advanced maternal age, ethnicity, and family history of type 2 diabetes mellitus [2]. Maternal obesity independently contributes to the development of GDM [3]. The Center of Disease Control (CDC) estimates that the incidence of GDM in the United States (US) is about 10%. It is reported to be higher in some countries with rates as high as 17.8–41.9% when using the International Association of Diabetes in Pregnancy Study Groups (IADPSG) GDM criteria [4,5]. In Western countries, a body mass index (BMI) of <25 Kg/m^2^ is considered normal. 

The recommendations for pharmacotherapy in patients with GDM have evolved over the past two decades. In this review, we provide an overview of the condition, with emphasis on the evolution of medical management over the past century. We have included the recent data on the options for medical management, as well as data on the current recommendations based on safety data of these pharmacotherapeutic options during pregnancy. 

## 2. Methods

This review was performed by including studies published in PubMed, Cochrane Library, the national guidelines, and WHO-based guidelines. The search was performed using key words that included “gestational diabetes”, “diabetes management”, and “pregnancy”. Recent publications, randomized control trials, observational studies, review articles, meta-analyses, Cochrane reviews, and current practice guidelines from national organizations were selected. Year of publication was not a limiting factor in this literature search. The publications included in this review were restricted to English language. The diagnosis of GDM in this review is based on the definition recommended by the American College of Obstetrics and Gynecology (ACOG). Articles on pharmacotherapy within the last two decades for treatment of hyperglycemia in GDM are reviewed with emphasis on the recent publications and recommendations. 

## 3. Historical Aspects of Gestational Diabetes

Recognition of diabetes complicating pregnancy dates back to 1873 [6]. In 1910, it was proposed that patients with gestational glycosuria be grouped into two categories, namely, pregnant women suffering from true and persistent glycosuria and pregnant women who pass sugar in their urine only when their diet contains a large amount of sugar or starch [7]. Subsequently, in a series of 468 pregnant women, 5 were documented with glycosuria with only one who was known to have pre-pregnancy diabetes [8]. The author suggested that in cases where glycosuria occurred during pregnancy, true diabetes might be “about to manifest itself”. From these early cases, it was recognized that there is a difference between diabetes diagnosed prior to pregnancy and diabetes diagnosed during pregnancy. 

Due to recognition of the possibility of diabetes affecting a pregnancy, studies were conducted to understand the effect of pregnancy on carbohydrate metabolism. The adverse effect of pregnancy on carbohydrate metabolism was first documented in 1946 [9]. Hoet from Belgium performed animal experiments to study carbohydrate metabolism during pregnancy [10]. The harmful effects of GDM on pregnancy outcomes were subsequently reported. In 1946, while reporting on the effects of hyperglycemia during pregnancy, the author documented the finding that fetal death can occur even before the woman has symptoms of diabetes [11]. Subsequently, in 1959, the association of maternal pre-diabetes with fetal macrosomia was published in Italy [12]. Two years later, O’Sullivan described GDM as “unsuspected and asymptomatic” diabetes, which is elicited in response to a glucose tolerance test during pregnancy [13]. At that time, the incidence of GDM was reported to be 1 in 116 [13].

Due to the recognition of the complications associated with GDM, a need for accurate and timely diagnosis was realized [14,15]. In 1924, a 50-g glucose load was administered to a pregnant patient with history of glycosuria, to determine if she had glucose intolerance [16]. The oral glucose tolerance test (GTT) was described in 1932. In search of an ideal glucose screen, an intravenous glucose tolerance test and a prednisone-glucose tolerance test were studied [17,18]. In 1961, O’Sullivan screened all pregnant patients with a 50-g oral glucose load and when the one-hour venous blood sugar was ≥130 mg/dL, the patient was considered screen-positive. At that time, patients were also classified as screen-positive if the patient had a family history of diabetes, a prior fetal or neonatal death, congenital fetal anomaly, prematurity, toxemia in two or more pregnancies, or a history of a prior baby that weighed 9 pounds or greater. Patients screened as positive were administered a three-hour GTT [13]. 

By 1985, glucose screening was routinely being used in patients considered to be at “high risk” for GDM [19]. Due to the varying definitions, the reported incidence of GDM ranged from 0.31% to 18%. This led to the suggestion of using similar screening criteria by all centers, to help better understand the condition and its impact on pregnancy.

## 4. Antenatal Glucose Screening

Various organizations attempted to establish population-based protocols to diagnose GDM [20]. In 1999, the WHO recommended a screening test of a 75-g anhydrous glucose load, following an overnight fasting for 8–14 h, between 24 and 28 weeks’ gestation. The protocols suggested by national and international organizations such as the IADPSG, American Diabetes Association (ADA), and United Kingdom-based National Institute for Health and Care Excellence (NICE) and Canada were recently reviewed [21]. Both a single-step and a two-step screening process during pregnancy have been described. Based on the Hyperglycemia and Adverse Pregnancy Outcome (HAPO) study, the IADPSG recommended a single-step 2-h glucose tolerance test. After an overnight fast, a 75-g glucose load is administered. The fasting 1-h and 2-h plasma glucose levels are checked. 

Despite the ADA endorsing the one-step 75-g glucose load GTT in 2011 based on the IADPSG criteria, the National Institute of Health in the USA recommended the two-step screening for GDM, which has been adopted by the American College of Obstetrics and Gynecology (ACOG) [22,23]. In the two-step approach, women first undergo a 1-h 50-g glucose screen, and if it is abnormal, a 3-h GTT with a 100-g glucose load is performed. If results are abnormal on the 3-h GTT, the patient is diagnosed with GDM. Based on the one-step IADPSG screening criteria, 17.8% of pregnant patients in the USA would test positive for GDM, which would nearly double the incidence of GDM in the USA [24]. The ACOG and other organizations in the USA have not adopted the one-step process due to a lack of evidence of impact on the pregnancy outcomes. The two-step testing at 24–28 weeks’ gestation starts with an initial screen in a non-fasting state, with an oral 50-g glucose load followed by a 1-h plasma glucose level. The cutoff value for the 1-h glucose screen is 130 to 140 mg/dL. Screen-positive women undergo the 3-h oral GTT. Following an overnight fast, a 100-g oral glucose load is administered. Plasma glucose levels are checked in the fasting state and at 1 h, 2 h, and 3 h following the glucose load [22]. Details of the different screening protocols are documented in Table 1. 

Early glucose screening is normally completed at the first prenatal visit in women with risk factors that include obesity with a BMI of ≥30 Kg/m^2^, history of gestational diabetes in a prior pregnancy, known impaired glucose metabolism, hemoglobin A1C of ≥5.7%, first-degree relative with diabetes mellitus, high-risk ethnicity, history of polycystic ovarian syndrome, pre-existing hypertension or cardiovascular disease, or a prior large baby ≥4000 g [22]. The early screening helps detect patients with pre-pregnancy type II diabetes mellitus. Women who have a normal glucose screen in early pregnancy have the test repeated at 24–28 weeks’ gestation.

Plasma glucose levels drawn in a fasting state, followed by a glucose load. The normal levels are as noted in the table. In the IADPSG, the Canada Diabetes Association, and the NICE guidelines, GDM is diagnosed when one or more plasma glucose levels are elevated above the normal levels. In the ACOG guidelines, patients with a positive 1-h glucose screen undergo the 3-h test and when two or more levels are elevated above the normal levels, GDM is diagnosed.

## 5. Pathophysiology of Gestational Diabetes

Claes Hellerström has been credited for the early work starting in 1963 on pancreatic changes during pregnancy and lactation in a mouse model (25,28). Insulin requirements physiologically increase during pregnancy. The increase in insulin demand is due to increased maternal caloric intake, maternal weight gain, presence of the placental hormones such as placental growth hormone, and placental lactogen, as well as increased prolactin and growth hormone production. As the pregnancy advances, the pancreatic β-cell mass increases to keep up with the demand for increased insulin. Failure of the β-cell expansion with a relative inadequate rise in insulin secretion leads to GDM. 

Maternal glucose is transported across the placenta to the fetus, and this delivery depends on the concentration gradient between the fetus and the maternal glucose levels. In the later part of pregnancy, the fetus diverts an increasing amount of maternal glucose towards itself, which leads to a decrease in maternal glucose levels. In order to maintain the concentration gradient of glucose across the placenta between the mother and the fetus, the maternal insulin resistance increases, as well as the hepatic glucose production [25]. In turn, the β-cells increase insulin secretion to prevent excessive delivery of glucose to the fetus. 

## 6. Management of Gestational Diabetes

The approach to optimal management of a patient diagnosed with GDM requires a multidisciplinary approach. This includes teaching patient self-monitoring of blood glucose levels, dietary modifications and nutrition monitoring, lifestyle changes, and maternal weight gain management. Up to 70–85% of patients diagnosed with gestational diabetes can be managed with adequate physical activity, and dietary and lifestyle modifications [26]. In 15–30% of patients, medications will be required. These medications include insulin as well as oral hypoglycemic agents (Figure 1). 

## 7. Blood Glucose Monitoring

Most organizations recommend daily self-glucose monitoring at home. Currently, the recommendations include daily self-monitoring with fasting and postprandial blood glucose. The ADA recommends the following target values: a fasting blood glucose of <95 mg/dL and a one-hour postprandial blood glucose of <140 mg/dL or a two-hour postprandial blood glucose of <120 mg/dL. Pre-prandial glucose monitoring is primarily for those with pre-existing diabetes. Monitoring with hemoglobin A1C levels is not as valuable for assessing glucose control in GDM [27]. Studies have examined alternative testing approaches to self-monitoring, such as healthcare-based monitoring and continuous glucose monitoring. A Cochrane review reported that there were no differences in self-monitoring compared with healthcare-based glucose monitoring for both maternal and neonatal complications. There were also no differences between self-monitoring and continuous glucose monitoring in cesarean section rates, large for gestational age, or neonatal hypoglycemia [28]. 

## 8. Dietary Interventions

Nutritional therapy must be discussed with a patient and, ideally, a diabetes educator counsels the patient regarding dietary modifications. General guidance in pregnancy is for women to consume three meals and two snacks during the day. Meal plans can be developed between the patient and the dietary educator to meet the appropriate daily requirements for pregnant diabetic patients, while trying to incorporate foods that are enjoyed by the patient as well feasible for the patient to follow. In conjunction with diet, women should be advised to write down their meals with their blood glucose values to help them identify foods that may contribute to postprandial hyperglycemia.

There are a variety of dietary approaches that have been described in the literature, including calorie-restricted diets, low-glycemic index diets, the DASH diet (dietary approaches to stop hypertension), low-carbohydrate diets, and low-unsaturated fat diets, high-fiber diets, and soy-based diets. Most of the dietary studies in pregnancy are reported to be of low-quality evidence [29]. ACOG references a 1993 study suggesting that the daily calorie allotment for gestational diabetics be distributed by percentages between the macronutrients with 40% from carbohydrates, 20% protein, and 40% fat [22]. However, the ADA’s 2017 review on diabetes care recommended similar dietary guidance for gestational diabetics to women with pre-existing diabetes. Fat consumption guidelines, cited from the Institute of Medicine, state that 20–35% of calories should be from fat. Higher-quality complex carbohydrates with lower glycemic indexes are preferred since they may help reduce the need for insulin as well as decrease postprandial hyperglycemia [30]. High-protein diets have not been demonstrated to improve health or glycemic control, and some studies suggest that excessive protein intake is associated with low fetal birthweight [30]. The ADA recommends daily protein requirements of 1–1.5 g/kg or 15–20% of consumed calories. Based on the above resources, dietary education should emphasize a balanced diet with portion control, healthy fats, complex carbohydrates, and 20% protein. 

## 9. Physical Activity

Exercise and physical activity are supported and encouraged during pregnancy, including in women with GDM. Moderate exercise during pregnancy has many beneficial effects including lower risk for development of GDM, lower likelihood of large for gestational age neonates, and being associated with lower hypertensive disorders and preterm birth, without an increase in fetal growth restriction [31,32]. Additionally, lifestyle changes during pregnancy have an influence on the postpartum period by lowering the risk of postpartum depression [33]. Current recommendations are for 30 min of moderate-intensity exercise, 5 days a week. When unable to perform moderate exercises, women may consider light exercises such as post-meal walks for 10–15 min, which can have a beneficial impact on blood sugar control. Pregnant women should avoid high-impact activities since they could result in abdominal trauma. 

## 10. Pharmacotherapy for Management of Gestational Diabetes

In up to 15–30% of patients with GDM, despite recommended diet and lifestyle modifications, blood glucose control is inadequate, and pharmacotherapy is required [34]. There are no well-studied protocols to determine the optimal time to initiate pharmacotherapy for glycemic control. Typically, after a period of 10–14 days of dietary and lifestyle modifications, if hyperglycemia persists through the day, pharmacotherapy should be considered. Based on a multicenter randomized trial by the NICHD, if the majority of fasting or postprandial blood glucose measurements are elevated or if there is suspicion of hyperglycemia with a random blood glucose level of ≥160 mg/dL or a fasting level of ≥95 mg/dL, medications should be initiated [35]. 

Insulin as well as oral medications has been used for the management of hyperglycemia in GDM patients. During pregnancy, insulin has the safest profile. The oral agents that have been studied include sulfonylurea such as glyburide (also known as glibenclamide), as well as metformin. In the USA, both the ADA and ACOG recommend insulin as the first line for control of hyperglycemia in patients with GDM. Insulin is a large molecule and does not cross the placenta. Metformin and glyburide have been shown to cross the placenta and into the fetus. 

## 11. Insulin Regimens

In patients that require insulin, the dose and timing of administration depend on the patient’s body weight, gestational age, and the time of day at which hyperglycemia is occurring. Once initiated, insulin doses are adjusted frequently throughout the pregnancy based on blood glucose results, symptomatic hypoglycemia, physical activity, dietary consumption, infection, and compliance. 

Insulin was discovered in 1922. In 1946, the intermediate-acting NPH insulin was introduced. For a long period of time, NPH was being used. Since then, research in different types of available insulin has evolved significantly. The insulin analogs that are currently available include rapidly acting analogs, such as aspart (Novolog) and lispro (Humalog), short-acting regular insulin, intermediate-acting NPH insulin, or longer-acting insulin analogs such as glargine (Lantus) and detemir (Levemir). In GDM, short-acting insulin is reported to increase the possibility of hypoglycemia and may cause fluctuations in glycemic control. Recent experience with aspart has been reassuring, while lispro has been associated with higher birth weight and increased rates of large for gestational age neonates [36]. For intermediate- and longer-acting insulin, randomized trials comparing detemir to NPH revealed no difference between the two in regard to glucose control and perinatal outcomes. Detemir has been associated with a lower incidence of hypoglycemia in the non-pregnant diabetic [37].

Two main approaches to prescribing insulin are based on the specific timing of recurrent hyperglycemia. Insulin can be administered throughout the day in divided doses or be given as a single daily dose depending on the timing of hyperglycemia. In women experiencing hyperglycemia only in the morning fasting state, intermediate insulin, such as NPH, or detemir should be administered at bedtime, as a single dose. In women experiencing postprandial hyperglycemia following specific meals only, rapid-acting insulin should be considered prior to the meal. Women experiencing hyperglycemia throughout the day should be managed with a combination of intermediate- or long-acting and short-acting insulin, with the total daily dose of insulin of 0.7–1.0 unit/Kg divided into rapid-acting insulin given before meals and intermediate- or long-acting insulin in the morning or at bedtime [22]. When prescribing insulin, close blood glucose monitoring is needed to try to avoid periods of hypoglycemia or hyperglycemia. Patients are advised to bring their self-monitored blood glucose logs to the office so the provider can determine if changes in the insulin regimen are needed. 

## 12. Oral Hypoglycemic Agents

Oral medications have not been adequately studied for possible long-term effects on neonatal outcomes, and therefore they are not recommended as the first choice in treatment for persistent hyperglycemia in GDM patients. These are also reported to fail in controlling hyperglycemia in about a quarter of women with GDM. In instances where patients are unable to get insulin, or refuse insulin, oral medications may be prescribed. When comparing the two oral agents, metformin appears to be safer than glyburide. Glyburide has been associated with neonatal hypoglycemia and higher birth weight, which can increase the risk for shoulder dystocia and need for cesarean delivery [38,39]. Additionally, glyburide is demonstrated to be present in umbilical cord blood samples in concentrations that are 50–70% those of maternal levels which can lead to neonatal hypoglycemia [38]. Even though metformin is stated to have a lower risk of neonatal hypoglycemia, the umbilical cord blood levels of metformin are reported to be similar to or higher than maternal levels [34]. 

Metformin dose starts at 500 mg orally each night or 500 mg twice daily based on glycemic control. Maximum total daily dose is 2500–3000 mg during pregnancy, which is higher than in the non-pregnant state. Glyburide is started at 2.5 mg daily or every 12 h and gradually increased to a maximum of 10 mg twice daily, based on glycemic control [22]. In pregnancy, the serum concentration of glyburide rises about 30–60 min after oral administration and peaks in 2–3 h, with the peak concentration coinciding with the peak of blood glucose following a meal. Since glyburide peaks at 2–3 h after administration, higher blood glucose values may be observed when checked 1 or 2 h following a meal, with a decrease subsequent to that. This may lead to increasing the glyburide dose but with a negative subsequent effect of hypoglycemia since the peak activity is 2–3 h following a meal [40]. When optimal glucose control on oral agents cannot be achieved, the option of insulin should be re-addressed with the patient. 

## 13. Maternal and Fetal Complications

Gestational diabetes has a strong impact on maternal and fetal outcomes. Women with GDM are at increased risk for development of pre-eclampsia and increased need for delivery by cesarean section. Much of this risk is related to the degree of glycemic control during the pregnancy, with poor glycemic control leading to higher risk for poor obstetrical outcomes [41]. These risks include obstetrical and neonatal complications such as preterm delivery, polyhydramnios, macrosomia, shoulder dystocia, admission to the neonatal intensive care unit, neonatal respiratory distress syndrome, fetal hypoglycemia, and hyperbilirubinemia [42]. There is also an increase in the risk of stillbirth. Aside from pregnancy-related risks, women diagnosed with gestational diabetes are at increased risk of developing type 2 diabetes mellitus later in life. 

## 14. Antepartum Surveillance

The need for fetal assessment is primarily based on maternal glycemic control. In women with good glycemic control on diet alone, fetal testing prior to 40 weeks’ gestation is not indicated, unless there are other indications [22]. If the patient is not delivered by the estimated due date, fetal testing can be performed weekly or twice weekly beginning at 40 weeks’ gestation. 

In women requiring pharmacotherapy, antepartum fetal testing should be initiated by 32 weeks’ gestation, consisting of a biophysical profile and non-stress test. These tests are performed twice weekly, similar to women with pre-existing diabetes, and until delivery [43]. Presence of other maternal co-morbidities or development of hypertensive disorders may impact the frequency and timing of antepartum testing, with some patients requiring fetal monitoring earlier than 32 weeks [22]. There is no clear consensus on ultrasound examinations for fetal growth assessment, but it must be performed at least once in the third trimester and closer to delivery to assess fetal growth. In summary, antepartum fetal monitoring is based on glycemic control, the use of medical therapy, and the presence of additional risk factors. Each institution should develop guidelines that are feasible based on the resources and facilities at that institution. 

## 15. Delivery Planning

ACOG published guidelines outlining the timing of delivery in women with gestational diabetes, which is based on glycemic control. When GDM is well controlled with diet and exercise, delivery should be considered starting at 39 weeks. Women requiring medications and with good glucose control are delivered between 39 and 39-6/7 weeks’ gestation. There is less firm evidence on the timing of delivery in women with poorly controlled GDM, which can be considered between 37 and 38-6/7 weeks’ gestation, taking into account the presence of other risk factors. In women with poor glycemic control but reassuring fetal testing, delivery in the preterm period is not recommended. The presence of hypertensive disorders, other co-morbidities, and non-reassuring fetal testing will alter the delivery timing [22].

In women with GDM and an estimated fetal weight of ≥4500 g on an ultrasound exam, the recommendation for cesarean is not well established [44]. In GDM cases with suspected fetal macrosomia, along with being counseled regarding the risks of possible shoulder dystocia with vaginal delivery, benefits and risks of cesarean delivery should also be discussed [22,45]. The timing and anticipated route of delivery should be reviewed with a patient during the third trimester. 

## 16. Intrapartum Glucose Management

During labor, women who have required insulin for GDM may have their blood glucose levels checked every hour or less frequently, depending on the antepartum requirements. Women that are diet-controlled can have less frequent blood glucose monitoring during labor, about every 4 h. Based on the findings of a recent randomized controlled trial of GDM patients in labor, blood sugar levels in the range of 60–120 mg/dL are acceptable. In the trial, tight intrapartum blood glucose control with a maternal blood glucose levels of 70–100 mg/dL was compared to more liberal control of 60–120 mg/dL. It was found that there was no difference in neonatal glucose concentrations between the two groups [46]. Based on the results of the study, when the blood glucose levels during labor are >120 mg/dL, intrapartum insulin must be considered. 

The starting dose or rate of intrapartum insulin is generally based on the severity of the elevated blood glucose value. If an insulin infusion is started, hourly blood glucose levels should be checked with the insulin rate adjusted based on the hourly blood glucose level. Insulin is infused in a 5% Dextrose solution or normal saline 0.9% (NS) at 125 mL/hour. The 5% glucose can be exchanged with NS if the blood glucose level is persistently greater than 180 mg/dL. Insulin infusions can be used for women undergoing both vaginal and cesarean delivery; however, in patients undergoing cesarean section, a one-time subcutaneous insulin injection can be considered to achieve euglycemia at the time of delivery. This management is one of many options to ensure euglycemia in the mother prior to delivery. There are a variety of other protocols that have been described in the literature to optimize intrapartum glucose control and the insulin protocol used may vary by institution. 

## 17. Postpartum

Following delivery, insulin and oral agents should be discontinued. In the immediate postpartum period, a fasting blood glucose level can be checked to determine if the patient has persistent hyperglycemia. A fasting plasma glucose level of ≥126 mg/dL or postprandial level of ≥200 mg/dL confirm persistent hyperglycemia. These patients are recommended to continue the diet modifications and lifestyle changes and may need to be treated with pharmacological agents. Insulin or glyburide can be prescribed in the postpartum period even in breast-feeding mothers, without concern for neonatal side effects [47]. 

In women with GDM, the risk for subsequent development of type II diabetes mellitus is estimated to be 10 times the control population, at 16.15% when followed up for 10 years. Some studies show the risk to be as high as 60% [47,48]. Postpartum fasting blood glucose often fails to identify patients with impaired glucose tolerance and those with type 2 diabetes mellitus. The Fifth International Workshop on Gestational Diabetes recommends women complete a 75-g 2-h oral glucose tolerance test at 6 to 12 weeks after delivery [44]. A fasting plasma glucose level of <100 mg/dL and a 2-h post-load plasma glucose level of <140 mg/dL are considered normal. When the fasting plasma glucose level is 100–125 mg/dL or the 2-h post-load glucose level is 140–199 mg/dL, the patient is considered to have impaired glucose tolerance. When the fasting plasma glucose level is ≥126 mg/dL or the 2-h post-load level is ≥200 mg/dL, the patient is diagnosed with diabetes mellitus. Recent data support performing the 2-h glucose tolerance test in the immediate postpartum period while the patient is still hospitalized [49]. This immediate postpartum glucose screening has not become standardized yet and this approach should be individualized based on the patient population and resources in the community. In conclusion, postpartum women need to be reminded of the importance of postpartum follow-up for the development of type II diabetes mellitus.

There are limitations to this review. Due to the vast amount of literature published on the subject, by various countries, it is possible that some important publications may have not been included in this review. The review was limited to the management of gestational diabetes in the United States, and this may differ in other countries where the incidence of obesity may be lower and resources available to patients may be different.

## 18. Discussion

Based on the published literature, we have reviewed the history of GDM, screening options, and management, with focus on the management options for persistent hyperglycemia. We report an increasing incidence of diabetes and GDM in association with increasing incidence in maternal obesity. 

Prevalence of GDM in the US has increased from <1% in 1961 to about 10%, using the two-step GTT for screening. With the introduction of the one-step 2-h 75-g GTT recommended by the IADPSG, about 18% in the US would qualify as having GDM (13–15). Due to a lack of evidence of improvement in pregnancy outcomes based on the one-step GTT, pregnant patients in the US continue to be screened at 24–28 weeks with the two-step GTT as recommended by the ACOG. Patients not previously diagnosed with diabetes mellitus but with risk factors for developing GDM are recommended to be screened earlier in gestation, and if negative, the screen is repeated at 24–28 weeks. 

Patients with GDM are followed closely with self-monitoring of blood glucose levels, and lifestyle changes which include modification in diet and exercise. In those with persistent hyperglycemia despite lifestyle changes, treatment with medications must be considered. Due to potential maternal and perinatal side effects of the oral hypoglycemic agents, insulin is the preferred treatment. Insulin regimens consist of short-, intermediate-, and long-acting insulin. The type, dose, and timing of insulin are determined by the timing and severity of hyperglycemia, and treatment is customized for each patient. Patients are followed closely, and the dose of insulin is adjusted at regular intervals based on the patient’s blood glucose levels.

In patients unable to take insulin, oral hypoglycemic agents can be considered. Metformin is preferred over glyburide due to the risk of possible neonatal hypoglycemia associated with maternal administration of glyburide. Typically, the dose of oral medications is once or twice a day, with increments in doses based on the degree, timing, and persistence of hyperglycemia. 

In patients requiring medications for persistent hyperglycemia, due to increased perinatal risks, fetal wellbeing tests are initiated by 32 weeks’ gestation. In patients that are well-controlled GDM on medications, delivery should be planned at 39 weeks and not earlier. However, in cases of maternal complications, poorly controlled hyperglycemia, or non-assuring fetal testing, the delivery may be considered earlier. 

Due to the increased incidence of impaired glucose tolerance or type 2 diabetes mellitus in patients with GDM, patient must be screened for diabetes in the postpartum period using the 2-h GTT. Long-term risk for development of type 2 diabetes mellitus is high. For early diagnosis of type 2 DM and to prevent long-term effects of diabetes mellitus, regular follow-up exams and testing, at a minimum of every 1–3 years, are recommended. 

Among high-risk patients that test positive for GDM on early GTT, some may indeed have type 2 diabetes mellitus that was previously undiagnosed. The ideal screening test in this population is not known as well as how early it should be administered. Patients diagnosed with GDM prior to 20 weeks are frequently managed similar to those with type 2 diabetes mellitus, but data on the pregnancy outcomes and perinatal outcomes in this group have not been well studied. Future studies on pregnancy outcomes in this population will become increasingly important, as the maternal obesity rates continue to rise. Whether obesity independently compounds the pregnancy risks in patients diagnosed with early onset of hyperglycemia is also an area of research.

## Figures and Tables

**Figure 1 ijerph-17-09573-f001:**
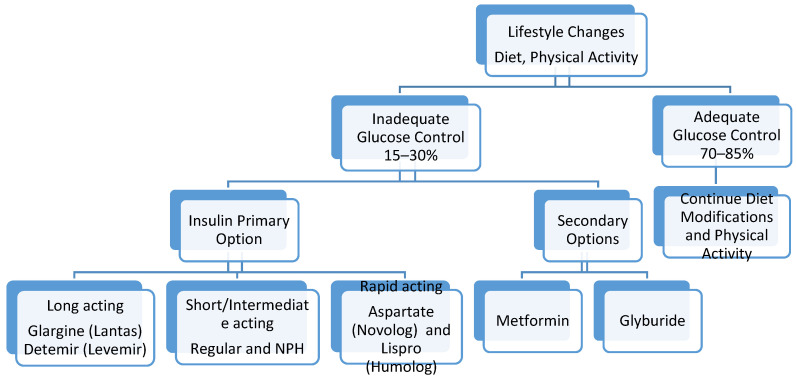
Management of hyperglycemia in gestational diabetes.

**Table 1 ijerph-17-09573-t001:** Protocol-based guidelines for diagnosis of gestational diabetes on oral glucose tolerance test.

Guidelines	Gestational Age at Screening	Glucose Load	FBS	1 h	2 h	3 h
IADPSG 2010	24–28 weeks	75 g	<92 mg/dL	<180 mg/dL	<153 mg/dL	-
Canada Diabetes Association 2018	24–28 weeks	75 g	<95 mg/dL <5.3 mmol/L	<190 mg/dL <10.6 mmol/L	<162 mg/dL <9.0 mmol/L	-
NICE 2015	24–28 weeks	75 g	<101 mg/dL <5.6 mmol/L		<140 mg/dL <7.8 mmol/L	-
ACOG 2018	24–28 weeks	100 g	<95 mg/dL	<180 mg/dL	<155 mg/dL	<140 mg/dL

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
