# Peer review of "Gestational Diabetes: Overview with Emphasis on Medical Management"

_ijerph, 2020, doi:10.3390/ijerph17249573_

Round 1

Reviewer 1 Report

Dear authors:

Thank you for the opportunity for reviewing your manuscript. However, I felt that reading this manuscript is hard to follow what you want to target on your update. I believe, if you be more focused and clear list your take-home message, it will be much more easier for readers. I have noticed a couple of questions that I’ve regarding your manuscript. (in special to my field of work which lifestyle behaviors). This section should be better framed and how would you approach the population with GDM?

Abstract: This section is incomplete, lacking important information, i.e., purpose, results encounter on your update (or review), and future directions/conclusion.

Introduction: It’s hard to follow the idea of this manuscript. I believe this section should be better structured to emphasize what news is being added to literature? In essence, the authors are providing the history of GDM… but this does not tell what the authors want to target, moreover what is the purpose of our study? How did you conduct this? Or are you aiming to provide a position paper on this topic?

Dietary interventions: this section should be better structured. Not only emphasizing energy and nutrients amounts but also how to change diet behaviors in these populations? How to target these recommendations. My comments are the same for the physical activity section? What it is new in terms of changing these behaviors? How can we approach them?

Are pharmacotherapy and oral agents not the same? Is metformin not a pharmacotherapy? So, why they on separate items?

Figure 1 is mentioned in the pharmacotherapy for GD? But this figure covers all the treatments/preventions for GD? Maybe should be explained as a summary of all prevention or treatments for GDM? Being that lifestyle behavior (diet + PA) do not treat, but can prevent. By the way, I really like this flowchart and this should be part of your review.

Author Response

Abstract: This section is incomplete, lacking important information, i.e., purpose, results encounter on your update (or review), and future directions/conclusion.

We have modified the Abstract and all of the above are included

Introduction: It’s hard to follow the idea of this manuscript. I believe this section should be better structured to emphasize what news is being added to literature? In essence, the authors are providing the history of GDM… but this does not tell what the authors want to target, moreover what is the purpose of our study? How did you conduct this? Or are you aiming to provide a position paper on this topic?

The Introduction has been modified and we have added a paragraph to describe the aim of this review. The entire manuscript has been modified to make it easy to read.

Dietary interventions: this section should be better structured. Not only emphasizing energy and nutrients amounts but also how to change diet behaviors in these populations? How to target these recommendations. My comments are the same for the physical activity section? What it is new in terms of changing these behaviors? How can we approach them?

There is mention of the dietary and lifestyle changes, but the review is focused on the medical management, which has been elaborated. The title of the paper has been changed to reflect the focus of the paper which is on Pharmacotherapy of Gestational Diabetes

Are pharmacotherapy and oral agents not the same? Is metformin not a pharmacotherapy? So, why they on separate items?

Pharmacotherapy now has two separate sections: Insulin and Oral agents

Figure 1 is mentioned in the pharmacotherapy for GD? But this figure covers all the treatments/preventions for GD? Maybe should be explained as a summary of all prevention or treatments for GDM? Being that lifestyle behavior (diet + PA) do not treat, but can prevent. By the way, I really like this flowchart and this should be part of your review.

In the text, Figure 1 is now mentioned at the start of the management. The details on dietary modifications, have been shortened, since it is not the focus of the paper

Reviewer 2 Report

An abstract prepares readers to follow the detailed information, analyses, and arguments in your full paper and should be accurately reflect the manuscript as a whole, in particular, the findings. Its need to be rewritten with balanced word distribution. Background section should be very briefly outline what the study intended to examine (or what the paper seeks to present). You may wish to add methods section adding such as how literature search was performed and indicates what sources have been checked to ensure a comprehensive assessment of relevant studies. Add summary of review findings.

The introduction provides a good, generalized background of the topic. The authors have included explanation of the topic and then provide context, and explain what are being extended to make the introduction more substantial. However, authors did not clearly include the review objectives. Authors may wish to add purpose of review perhaps in the last couple of lines of introduction.

Author did not provide information about Methods. The review states how literature search was performed and indicates what sources have been checked to ensure a comprehensive assessment of relevant studies (e.g., MEDLINE, the Cochrane Collaboration Database, the Centre for Research Support, PubMed).

Author Response

An abstract prepares readers to follow the detailed information, analyses, and arguments in your full paper and should be accurately reflect the manuscript as a whole, in particular, the findings. Its need to be rewritten with balanced word distribution. Background section should be very briefly outline what the study intended to examine (or what the paper seeks to present). You may wish to add methods section adding such as how literature search was performed and indicates what sources have been checked to ensure a comprehensive assessment of relevant studies. Add summary of review findings.

The manuscript has been re-written, several sentences eliminated, making the flow easier. The background, which states the purpose of the paper. A method section has been added.

The introduction provides a good, generalized background of the topic. The authors have included explanation of the topic and then provide context, and explain what are being extended to make the introduction more substantial. However, authors did not clearly include the review objectives. Authors may wish to add purpose of review perhaps in the last couple of lines of introduction.

The abstract has been modified, to include the purpose for the review.

Author did not provide information about Methods. The review states how literature search was performed and indicates what sources have been checked to ensure a comprehensive assessment of relevant studies (e.g., MEDLINE, the Cochrane Collaboration Database, the Centre for Research Support, PubMed).

A method section has been added

Reviewer 3 Report

This review tried to summarize the update in gestational diabetes. The title of this manuscript suggests a huge range of the content about gestational diabetes. However, the authors mainly focused on the clinical management of gestational diabetes. The section "Pathophysiology of gestational diabetes" is not informative, lacking the updates in this research field. The authors should provide up-to-date progress in the mechanism underlying this disease, such as β-cell dysfunction,  insulin resistance, oxidative stress, protein aggregation, etc. Otherwise, the title should be tailored to fit the focus that the authors currently wrote.

Author Response

This review tried to summarize the update in gestational diabetes. The title of this manuscript suggests a huge range of the content about gestational diabetes. However, the authors mainly focused on the clinical management of gestational diabetes. The section "Pathophysiology of gestational diabetes" is not informative, lacking the updates in this research field. The authors should provide up-to-date progress in the mechanism underlying this disease, such as β-cell dysfunction,  insulin resistance, oxidative stress, protein aggregation, etc. Otherwise, the title should be tailored to fit the focus that the authors currently wrote.

The title of the paper has been changed, to focus on the Medical management of gestational diabetes. Due to the vast amount of published literature on the subject of gestational diabetes, the contents have been revised to adhere to the theme of the review – Management of Gestational Diabetes, in the US, which was the topic that was assigned to us.

Round 2

Reviewer 3 Report

The authors have addressed my concerns and significantly improved the manuscript. Now it is acceptable for publication.

Author Response

A separate comment section by Reviewer 3 was not seen. Below are the replies to the comments made by the Academic Editor:

To improve text flow after the Postpartum subsection the author should include a heading named Discussion:

A Discussion section has been added.

The format of the summary looks more like a report rather a scientific manuscript:
The Summary part has been deleted and information has been incorporated in the Discussion section.

What is already known about the topic? and What are its contributions to the literature and body of evidence?
Since this is [an invited] review paper, the Discussion clarifies that it is a Review paper and that the evidence regarding the medical management is the focus.

The summary topic could be presented as a running text subdivided into sections related to recommendations for research and recommendations for clinical practice and public health policy:
The summary has been deleted, and the information is in a text form, and incorporated in the Discussion section.
Recommendations for research and clinical practice has been added to the Discussion section.